# Hydrologic Mass Changes and Their Implications in Mediterranean-Climate Turkey from GRACE Measurements

**Gonca Okay Ahi [1],*** and **Shuanggen Jin [2,3],***

[1]   Department of Geomatics Engineering, Hacettepe University, Beytepe Campus, 06800 Ankara, Turkey
[2]   Shanghai Astronomical Observatory, Chinese Academy of Sciences, Shanghai 200030, China
[3]   School of Remote Sensing and Geomatics Engineering, Nanjing University of Information Science and Technology, Nanjing 210044, China
*   Correspondence: goncaokayahi@hacettepe.edu.tr (G.O.A.); sgjin@shao.ac.cn (S.J.);
    Tel.: +90-312-297-6990 (G.O.A.)

**Abstract:** Water is arguably our most precious resource, which is related to the hydrological cycle, climate change, regional drought events, and water resource management. In Turkey, besides traditional hydrological studies, Terrestrial Water Storage (TWS) is poorly investigated at a continental scale, with limited and sparse observations. Moreover, TWS is a key parameter for studying drought events through the analysis of its variation. In this paper, TWS variation, and thus drought analysis, spatial mass distribution, long-term mass change, and impact on TWS variation from the parameter scale (e.g., precipitation, rainfall rate, evapotranspiration, soil moisture) to the climatic change perspective are investigated. GRACE (Gravity Recovery and Climate Experiment) Level 3 (Release05-RL05) monthly land mass data of the Centre for Space Research (CSR) processing center covering the period from April 2002 to January 2016, Global Land Data Assimilation System (GLDAS: Mosaic (MOS), NOAH, Variable Infiltration Capacity (VIC)), and Tropical Rainfall Measuring Mission (TRMM-3B43) models and drought indices such as self-calibrating Palmer Drought Severity (SCPDSI), El Niño–Southern Oscillation (ENSO), and North Atlantic Oscillation (NAO) are used for this purpose. Turkey experienced serious drought events interpreted with a significant decrease in the TWS signal during the studied time period. GRACE can help to better predict the possible drought nine months before in terms of a decreasing trend compared to previous studies, which do not take satellite gravity data into account. Moreover, the GRACE signal is more sensitive to agricultural and hydrological drought compared to meteorological drought. Precipitation is an important parameter affecting the spatial pattern of the mass distribution and also the spatial change by inducing an acceleration signal from the eastern side to the western side. In Turkey, the La Nina effect probably has an important role in the meteorological drought turning into agricultural and hydrological drought.

**Keywords:** terrestrial water storage (TWS); GRACE; GLDAS; TRMM; drought; ENSO; NAO; Turkey

---

## 1. Introduction

Traditional methods of monitoring hydrological processes (e.g., in situ measurements of precipitation and soil moisture content) have generally been inadequate to characterize extreme hydrologic events [1,2]. Their temporal and spatial resolutions are not good enough to characterise water mass variations at a regional or global scale. In order to improve our knowledge to predict and monitor these water mass changes in the scope of drought analysis, flood potential assessment, groundwater changes, soil moisture analysis, etc., there are an increasing number of available datasets being produced, especially from remote sensing techniques. These techniques offer information

on vegetation, precipitation, surface water storage, evapotranspiration, soil moisture, groundwater, and snow components. Tropical Rainfall Measuring Mission (TRMM) [3], TRMM Multisatellite Precipitation Analysis (TMPA), Precipitation Estimation from Remotely Sensed Information using Artificial Neural Networks (PERSIANN) [4], the CPC MORPHing technique (CMORPH) [5], Climate Hazards Group InfraRed Precipitation with Station data (CHIRPS) [6], Global Satellite Mapping of Precipitation (GSMaP) [7–9], and Global Precipitation Measurement (GPM) [3] are some of the available satellite precipitation missions offering multiple products in real-time. While missions like Soil Moisture Active Passive (SMAP) provide satellite soil moisture values, the U.S. Department of Agriculture's (USDA) global reservoir and lake monitoring service provides information on satellite surface water levels by using near-real-time radar altimeter data. Finally, to study satellite surface/subsurface water, since 2002, the Gravity Recovery and Climate Experiment (GRACE) has provided entirely new observations suitable for quantifying and monitoring continental or regional TWS changes [10–13], groundwater changes [14,15], drought monitoring [16–19], and flood potential assessments [20] at a spatial resolution of a few hundred kilometers with uniform data coverage. The approaches used for estimating groundwater storage variations with the main applications of GRACE data for groundwater monitoring can be found in [21]. Moreover, since 2018, GRACE Follow-On (GRACE-FO) satellite gravity mission has been established to continue tracking Earth's water movements at different spatial scales. The results retrieved from the satellite gravity measurements are independent of the in situ data and might be interpreted independently. However, to increase the accuracy and make a better interpretation, the trend needs to be assessed based on a combination of additional data sets provided by other remote sensing technics. Moreover, as observed in many studies, the comparison of global hydrological models' results with GRACE data supports the analyses. The Global Land Data Assimilation System (GLDAS) [22], which is developed jointly by NASA and NOAA, simulates Terrestrial Water Storage through four main land surface models: VIC [23], NOAH [24], Mosaic [25], and CLM [26]. The Climate Prediction Center (CPC) model [27], Land Dynamics model (LadWorld) [28], WaterGAP Global Hydrology Model (WGHM) [29–31], and Organizing Carbon and Hydrology in Dynamic Ecosystems (ORCHIDEE) Land surface model [32] are examples of such global hydrological models [33], which provide a general overview on the use of global hydrological models' results (e.g., water storage change) as a reference to calibrate/validate GRACE data. Additionally, [33] found inconsistencies in the previous studies between hydrological model simulation results and GRACE-based observations and provided possible explanations for these inconsistencies.

In Turkey, besides traditional hydrological studies, TWS is poorly investigated at a continental scale with the new satellite techniques (e.g., GRACE). Previous studies in Turkey revealed about 0.7 cm/year the TWS variation [34]. Among them, the study of TWS variations in Turkey [35] with GRACE and GLDAS (NOAH) data sets [22] for the period from 2003 to 2009, showed a significant decrease of up to a rate of 4 cm/year for both data sets in the southern part of the central Anatolian region. This decrease present in both datasets is explained by decreasing groundwater variations confirmed by the existing well in the above mentioned regions. More recently, the effect of drought and water extraction on groundwater storage in central Turkey are also described [36]. They also showed how the groundwater storage can affect the TWS. In addition, long-term TWS changes in Turkey during the 2004–2014 period by associating GLDAS/NOAH data are studied and accounted for TWS variation between −17 and 16 cm in amplitude, with an important decrease in 2008 [37]. The requirement of further studies to isolate model errors and anthropogenic effects for Turkey, in order to explain the GRACE signal, which points out a robust acceleration in TWSA is emphasized also by other studies [38]. These prior studies point out the need for a summarizing and extended up-to-date study which accounts for a longer time period, associating different auxiliary data and parameters to understand and interpret the TWS change mechanism and possible drought events at a national scale.

In this paper, the water storage variation is studied over Turkey at a seasonal time scale for the period from April 2002 to January 2016 using monthly GRACE land mass grids (Level 3-RL05) from CSR. To compare the results, monthly grids of GLDAS data with a $1° \times 1°$ resolution and three GLDAS

hydrological models: MOS, NOAH, VIC, are also analyzed. To estimate precipitation over Turkey within the studied period, the TRMM-3B43 model is also considered, with a 0.25° × 0.25° resolution. In addition, ENSO, the self-calibrating Palmer Drought Severity Index, and NAO are compared with the satellite-derived GRACE TWS data.

## 2. Data and Methods

### 2.1. Studied Area and Its Hydrological Characteristics

Turkey is geographically located at approximately 36–42° N and 26–45° E, with an approximate area of 783.562 km$^2$. A glance at a digital elevation model of Turkey reveals that mountains encircle the peninsula of Anatolia in four directions (see Figure 1).

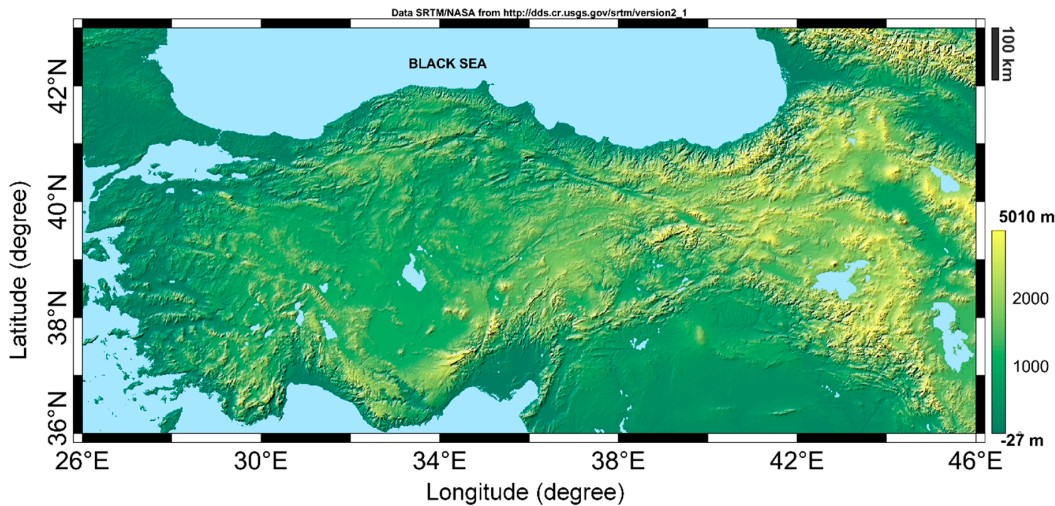

**Figure 1.** Digital Elevation Model of Turkey.

In Turkey, the yearly mean precipitation rate is approximately 643 mm (501 billion m$^3$ water). A total of 274 billion m$^3$ of that water is transferred to the atmosphere by evaporation from soil, water, and vegetation surfaces. Additionally, 158 billion m$^3$ flow to the sea and lakes in closed basins through streams. Furthermore, 28 billion m$^3$ out of the remaining 69 billion m$^3$ water feeding the groundwater contributes to the surface water. Besides, an additional water input of about 7 billion m$^3$ comes from neighboring countries. Thus, the surface water potential of the country is 193 billion m$^3$, or 234 billion m$^3$ by adding the 41 billion m$^3$ of water contributing to the groundwater [37,39].

Important and severe drought events experienced in Turkey are recorded between 1971–1974, 1983–1984, 1989–1990, 1996, 2001, and 2007–2008 [40,41]. There are also other studies investigating droughts in Turkey revealing an agricultural and hydrological drought starting from November/December 2006 to December 2008, so the drought period is recorded as 2007–2008 [42]. Especially in 2008, there was no variation in snow and precipitation for nine to 10 months [43].

### 2.2. Terrestrial Water Storage (TWS) from GRACE

Three processing centers, including CSR (Center for Space Research, Texas), JPL (Jet Propulsion Laboratory, California), and GFZ (GeoForschungsZentrum, Potsdam), provide official releases of GRACE gravity data at three different levels (Level 1, 2, 3), depending on the expertise and needs of the users for both the time-averaged and time-variable fields. In this study, Level 3 (RL05) land data of the CSR processing center have been used, which were ready to use as many necessary preprocessing steps had already been applied (removal of atmospheric pressure/mass changes, replacement of the C20 (degree 2 order 0) coefficients with the solutions from Satellite Laser Ranging [44], the estimation of the degree-1 coefficients (geocenter) from [45], the correction of glacial isostatic adjustment (GIA),

destriping [46], Gaussian filtering). Level 3 data are in the form of GRACE-derived mass grids expressed as the TWS function of the gravity fields with 1 degree in both latitude and longitude (approx. 111 km at the equator) spatial sampling and estimates over land from the gravity coefficient anomalies for each month ($\Delta C_{lm}$, $\Delta S_{lm}$) [47], as below:

$$\Delta \eta_{land}(\theta, \phi, t) = \frac{a\rho_{ave}}{3\rho_w} \sum_{l=0}^{\infty} \sum_{m=0}^{l} \widetilde{P}_{lm}(\cos\theta) \frac{2l+1}{1+k_l} (\Delta C_{lm} \cos(m\phi) + \Delta S_{lm} \sin(m\phi)) \tag{1}$$

where $\rho_{ave}$ is the average density of the Earth, $\rho_w$ is the density of fresh water, $a$ is the equatorial radius of the Earth, $\widetilde{P}_{lm}$ is the fully-normalized Legendre associated function of degree l and order m, $k_l$ is the Love number of degree l [48], $\theta$ is the spherical co-latitude (polar distance), and $\phi$ is the longitude. All grids are obtained from the following link: https://grace.jpl.nasa.gov/data/get-data/monthly-mass-grids-land/.

The first, additionally applied data processing, also recommended by the processing center for the Level 3 land data grid, in order to prevent possible attenuation of the surface mass variations due to the sampling and post-processing of GRACE observations (destriping, gaussian) and to regain part of the information loss in prior data processing, is the multiplication of one for each 1-degree land grid by a set of provided scaling coefficients, as shown below in Equation (2):

$$g'(x, y, t) = g(x, y, t) \times s(x, y) \tag{2}$$

where $x$ is the longitude index, $y$ is the latitude index, $t$ is time (month) index, $g(x, y, t)$ is the grid node, $s(x, y)$ is the scaling grid, and $g'(x, y, t)$ is the gain-corrected time series. Moreover, additionally applied data processing, leakage error correction (residual errors after filtering and rescaling), has been performed (as below in Equation (3)) with the provided file obtained from the following link: ftp://podaac-ftp.jpl.nasa.gov/allData/tellus/L3/land_mass/RL05/netcdf/, containing scaling coefficients (as mentioned previously) and leakage error estimates.

$$g'_{leak\_corr}(x, y, t) = g'(x, y, t) + leakage\_err(x, y) \tag{3}$$

where $x$ is the longitude index, $y$ is the latitude index, $t$ is the time (month) index, $g'(x, y, t)$ is the gain-corrected time series, *leakage_err*$(x, y)$ is the leakage error estimates, and $g'_{leak\_corr}(x, y, t)$ is the scaled and leakage error corrected time series. In this study, monthly mass grids of GRACE land data (Level-3 RL05) from the CSR processing center concerning the period from April 2002 to January 2016 with a $1° \times 1°$ spatial resolution have been used after applying these additional processing steps.

*2.3. Global Land Data Assimilation System (GLDAS) Models Data*

GLDAS has been developed jointly by scientists at the National Aeronautics and Space Administration (NASA)-Goddard Space Flight Center (GSFC) and the National Oceanic and Atmospheric Administration (NOAA)-National Centers for Environmental Prediction (NCEP). GLDAS is a global, high-resolution, offline (uncoupled to the atmosphere) terrestrial modeling system that incorporates satellite- and ground-based observations in order to produce optimal fields of land surface states (e.g., soil moisture, snow water equivalent, and canopy water storage...) and fluxes (e.g., rainfall, snowmelt, evapotranspiration...) in near–real time [22]. Currently, GLDAS drives four land surface models: MOS, NOAH, the Community Land Model (CLM), and the VIC. In this study, GLDAS version1 (GLDAS-1) monthly data of the four land surface models are downloaded from https://disc.gsfc.nasa.gov/datasets?keywords=gldas&page=1, with a $1° \times 1°$ spatial resolution, concerning the period from January 2002 to January 2016. In these data sets, the GLDAS provides a time series of land surface states and fluxes (25 variables), which can be used to study water storage. The anomalies corresponding to the major part of the signal to TWS can be assumed to arise from the change in soil moisture (kg/m$^2$), snow water equivalent (kg/m$^2$), and canopy water storage (kg/m$^2$).

Hence, firstly, these land surface state variables are derived from the file covering Turkey and then, the TWS from GLDAS models is calculated, as shown by Equation (4):

$$TWS_{GLDAS} = \Delta SM + \Delta SWE + \Delta CWS \tag{4}$$

where, $TWS_{GLDAS}$ is the change in terrestrial water storage from GLDAS, $\Delta SM$ is the change in soil moisture, $\Delta SWE$ is the change in the snow moisture equivalent, and $\Delta CWS$ is the change in canopy water storage. Soil moisture values are averaged before integrating them into the TWS calculation, according to the three-layer model for VIC and MOS, and the four-layer model for NOAH. In this study, CLM models have not been used.

### 2.4. Tropical Rainfall Measuring Mission (TRMM) Data

TRMM is a joint mission between NASA and the Japan Aerospace Exploration (JAXA) Agency in order to study rainfall for weather and climate research (1997–2015). With the help of several space-borne instruments, TRMM satellite data allow precipitation from diurnal to interannual time scales to be measured, which led to improving our understanding of tropical cyclone structure and evolution, including important variability associated with the Madden-Julian Oscillation and with El Nino Southern Oscillation (ENSO), convective system properties, lightning-storm relationships, climate and weather modeling, and human impacts on rainfall. The data also supported operational applications such as flood and drought monitoring and weather forecasting (https://trmm.gsfc.nasa.gov/).

In our study, we used the TRMM-3B43 Level 3 gridded monthly satellite-gauge (SG) combination data set with a 0.25° × 0.25° degree spatial resolution downloaded from http://mirador.gsfc.nasa.gov/# to estimate the precipitation variations over Turkey.

### 2.5. In-Situ Precipitation Data

The Turkish state meteorological service provides an annual cumulative rainfall distribution (1981–2010) map produced on a GIS platform by kriging the in-situ rainfall data of 255 meteorological stations. This map has been downloaded from the following website: https://mgm.gov.tr/eng/forecast-cities.aspx, and used further to compare/validate TRMM data and rainfall from ground meteorological stations.

### 2.6. Self-Calibrating Palmer Drought Severity Index (SCPDSI) Data

The SCPDSI [49] can estimate the departure relative to normal conditions in the surface water balance by using a hydrological accounting system [50,51]. The PDSI is primarily considered a meteorological drought indicator, and sometimes, an agricultural drought indicator [52]. The needed drought index data was downloaded from the following website: https://crudata.uea.ac.uk/cru/data/drought/, as global land data covering the time period from 1901 to 2016 with a 0.5° latitude-longitude spatial resolution. Then, grids corresponding to Turkey and the time period from 2002 to 2016 were extracted from global data.

### 2.7. El Niño–Southern Oscillation (ENSO) Index Data

ENSO is described as warming on the ocean surface, or above-average sea surface temperatures (SST), in the central and eastern tropical Pacific Ocean. This is one of the most important climate phenomena on Earth due to its ability to change the global atmospheric circulation, which in turn, influences temperature and precipitation across the globe. The magnitude of the ENSO is often expressed by the Niño SST3.4 index, derived from the normalized Sea Surface Temperature (SST). El Nino (warm phase) and La Niña (cold phase) are two contrary phases of ENSO [53]. In order to understand the magnitude of ENSO which influences precipitation and to improve our understanding of the occurrence of drought events at a national scale, SST data were downloaded from the following

website: http://www.cpc.ncep.noaa.gov/data/indices/, where monthly ERSSTv5 (1981–2010 base period) Niño 1 + 2 (0–10° South) (90° W–80° W) Niño 3 (5° N–5° S) (150° W–90° W) Niño 4 (5° N–5° S) (160° E–150° W) Niño 3.4 (5° N–5° S) (170 W–120° W) is available. The time period (from 2002 to 2016) was extracted from global data.

*2.8. North Atlantic Oscillation (NAO) Index Data*

NAO is the variability in atmospheric mass circulation especially observed in the cold season months (November–April) over the middle and high latitudes of the Northern Hemisphere (from central North America to Europe and much into Northern Asia). The understanding of its mechanism on the surface temperature, storms, precipitation, ocean, and ecosystem results in understanding global climate change [54]. Strong positive phases (+) of the NAO tend to be associated with below-average precipitation over southern and central Europe. Conversely, above average temperature and precipitation anomalies are typically observed during strong negative phases (-) of the NAO (https://www.cpc.ncep.noaa.gov/data/teledoc/nao.shtml). The monthly mean NAO index data from 2002 to 2017 were downloaded from the following website: https://www.cpc.ncep.noaa.gov/products/precip/CWlink/pna/nao.shtml.

To conclude the data and methods section and proceed with the results and analysis section, Table 1 below summarizes the studied research topics, used input data, methodology, and additional processing applied in this paper.

**Table 1.** Studied topics and general structure of the paper.

| Research Topic | 3.1. Drought Analysis from Time Series | 3.2. Spatial Mass Distribution and Its Causes | 3.3. Long-Term Mass Change | 3.4. Impact on TWS Variations | 3.5. Understanding Drought and Its Relation with Climatic Change |
|---|---|---|---|---|---|
| Input data (global) | GRACE and GLDAS TWS | GRACE, GLDAS TRMM, In-situ | GRACE and GLDAS TWS | GRACE(TWS), GLDAS **, TRMM (precipitation) | GRACE(TWS), SCPDSI, ENSO, NAO |
| Methodology | from the analyses of TWS time series | Harmonic analyses and least squares fitting according to [31] for both GRACE and GLDAS data Annual amplitude (cm), phase (degree) and trend (cm/yr) values are derived | from the analyses of TWS time series | Evapotranspiration(t) = P(t) − R(t) – TWS (GRACE) | from the analyses of TWS time series |
| Additional processing | -GRACE: Scaling and leakage error correction -GLDAS: soil moisture values are summed with respect to the number of models layers as following: three-layer model for MOS and VIC and four-layer model for NOAH TWS(GLDAS) = ΔSM + ΔSWE + ΔCWS * | * | * | GLDAS: soil moisture values are summed with respect to the number of models layers as following: three-layer model for MOS and VIC and four-layer model for NOAH * | * |

* Interested time (~2002–2016) and land grids (Turkey) are extracted for all global data sets. ** Rainfall rate, Evapotranspiration, Soil moisture. P(t) = Precipitation from TRMM. R(t) = Runoff derived from MOS, NOAH, and VIC models.

## 3. Results and Analysis

### 3.1. Drought Analysis from Time Series

In order to understand important drought periods, TWS time series derived from GRACE data and from GLDAS models were analyzed in the studied time period from 2002 to 2016. Figure 2 indicates residual (GRACE TWS-mean GRACE TWS) monthly TWS variations in Turkey (cm) according to GRACE and GLDAS models (MOS, NOAH, VIC).

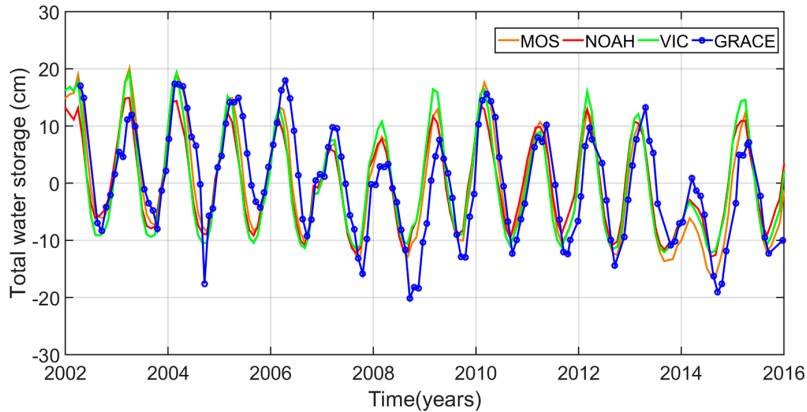

**Figure 2.** Residual monthly TWS variations in Turkey (cm) according to GRACE (GRACE TWS-mean GRACE TWS) (in blue with circle), GLDAS-MOS TWS—mean (GLDAS MOS TWS) (in orange), GLDAS NOAH TWS—mean (GLDAS NOAH TWS) (in red), and GLDAS VIC TWS—mean (GLDAS VIC TWS) (in green) models.

Figure 2 indicates that GRACE TWS time series and GLDAS models are consistent. According to the GRACE TWS signal, the studied time period shows some sudden TWS decreases in September 2004 and 2008, which falls after dry summer periods, and in October 2014. However, to be prudent, the decrease in September 2004 may be an over estimation of GRACE solutions. Additionally, there are some significant increasing and decreasing trends in TWS during some time intervals. Firstly, GRACE TWS time series indicate an important decreasing trend period from February 2006 to November 2008 [42]. This underlines the beginning of an agricultural and hydrological drought, which occurred due to the below-average precipitation [55] from November/December 2006 to December 2008. In this context, GRACE time series provide an earlier warning (~9 months before) for the beginning of a decreasing trend in TWS. Briefly, GRACE indicates that 2008 is a remarkable year in terms of drought records. This finding is also supported with no snow and precipitation variation in 2008 [43]. After this decreasing time interval, Turkey is exposed to an increasing trend in TWS from November 2008 to March 2010. According to Figure 2, the second decreasing period begins in March 2010 and lasts until October 2014. During this time interval, historical records indicate the beginning of a meteorological drought in 2012, intensified with dry summers, as is usual for the Mediterranean climate. Even in 2013, the observed above-average precipitation levels (32 cm in 2013 where average is 27 cm) could not stop the decreasing trend in TWS because of the successively observed below-average precipitation that occurred in 2014 (17 cm) [56]. To resume, the analysis of GRACE TWS time series and the results of previous works revealed that the GRACE signal is mostly sensitive to agricultural drought (insufficient soil moisture content) and hydrological drought (significant reduction in winter precipitation) (2006–2008) compared to meteorological drought (little rain combined with increased temperature and lower humidity) (2012–2014) for the studied region. Following this prior understanding about the important increasing and decreasing trends in TWS, leading to important drought events in the time period from 2002 to 2016, we now focus (Section 3.2) on the spatial mass distribution of the TWS and its causes at the national scale.

*3.2. Spatial Mass Distribution and Its Causes*

The TWS variations have significant seasonal signals. As we are interested in seasonal variations with significant annual and semiannual periods, a mathematical function/model (Equation (5)), which includes the annual and semi-annual variations with linear trend terms, has to be used to fit the TWS time series [57]:

$$M(t) = a + bt + \sum_{k=1}^{2} c_k \sin(\omega_k(t - t_0) + \phi_k) + \varepsilon(t) \tag{5}$$

where $M(t)$ is the time series; $t$ is the time; $t_0$ is a reference time; $a$ is the constant; $b$ is the trend; $c_k$, $\phi_k$, and $\omega_k$ are annual amplitude, phase, and frequency, respectively; k = 1 is for the annual variation and k = 2 is for the semi-annual variation; and $\varepsilon(t)$ is the un-modeled residual term. During the analysis of time series, strong annual signals are found, and k = 1 year is used here. Using the least-squares method to fit the time series of GRACE data at each point, the annual amplitude, annual phase, and trend terms of TWS are estimated. Figures 3 and 4 show the annual amplitude (cm) and annual phase (degree) map of TWS variation of Turkey from GRACE data and GLDAS models (MOS, NOAH, VIC) concerning the 2002–2016 period, respectively. The TWS variation estimates of both methods show a good agreement in annual amplitudes and phases. The mean annual amplitude is 11.06 cm and 11.19 cm from GRACE and GLDAS models (MOS), respectively. Additionally, the mean annual phase is 21.90 and 22.98° from GRACE and GLDAS models (MOS), respectively. The larger annual amplitude values >20 cm (in red, Figure 3a,d) are observed in the eastern parts of Turkey. However, significant amplitude values also appear at shorelines (also Black Sea, Aegean, Mediterranean). The smaller annual amplitude of TWS variations of nearly 2.89 cm is seen in the middle of Turkey (central Anatolia, Figure 3a–d). According to phase plots, there is lateral variation (especially concerning GRACE phase plot, Figure 4a) from eastern part to the western part of Turkey, with a small increase through to the west side contrary to the amplitude plots. This means that lower phase values are observed in the eastern parts of Turkey, where larger amplitude values have been previously estimated. Table 2 indicates the mean annual amplitude, phase, and trend variations in Turkey according to GRACE and GLDAS models.

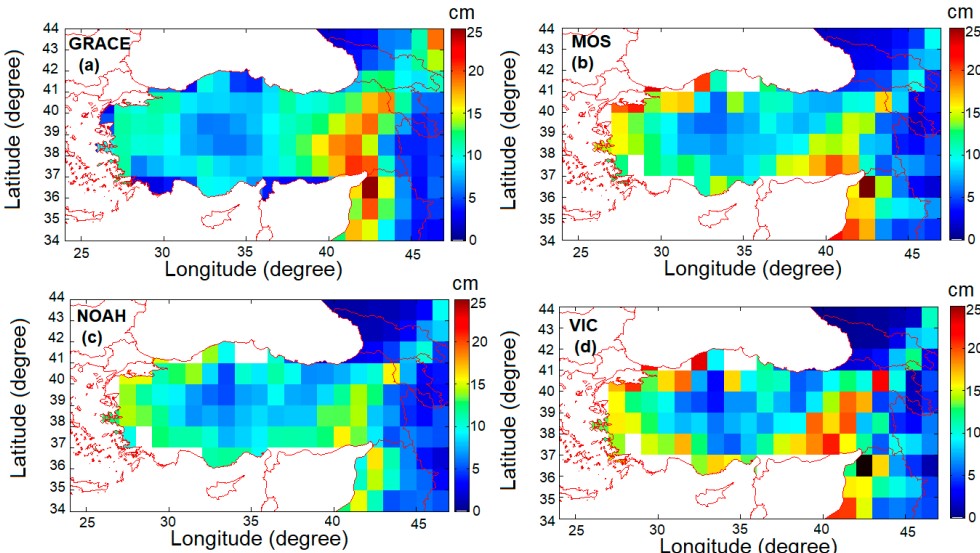

**Figure 3.** Annual amplitude (cm) of TWS variation of Turkey from (**a**) GRACE data and GLDAS models, (**b**) MOS, (**c**) NOAH, and (**d**) VIC concerning the 2002–2016 period.

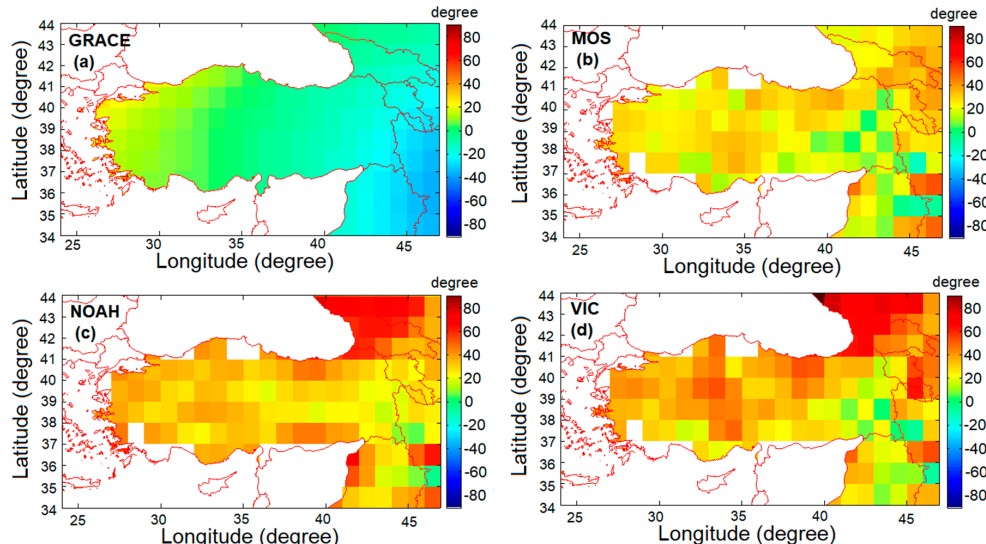

**Figure 4.** Annual phase (degree) of TWS estimated from (**a**) GRACE data and GLDAS models, (**b**) MOS, (**c**) NOAH, and (**d**) VIC concerning the 2002–2016 period. In order to see small variation, the color bar is limited.

**Table 2.** Mean annual amplitude, phase, and trend variations in Turkey according to GRACE and GLDAS models.

| Groundwater Storage | Annual Amplitude (cm) | Annual Phase (degree) | Trend (cm/yr) |
|---|---|---|---|
| GRACE | 11.06 | 21.90 | −0.77 |
| GLDAS-MOS | 11.19 | 22.98 | −0.74 |
| GLDAS-NOAH | 10.00 | 32.25 | −0.31 |
| GLDAS-VIC | 11.81 | 32.22 | −0.39 |

Finally, Figure 5 shows the trend (cm/yr) of TWS of Turkey according to GRACE and GLDAS models (MOS, NOAH, VIC) concerning the 2002–2016 period.

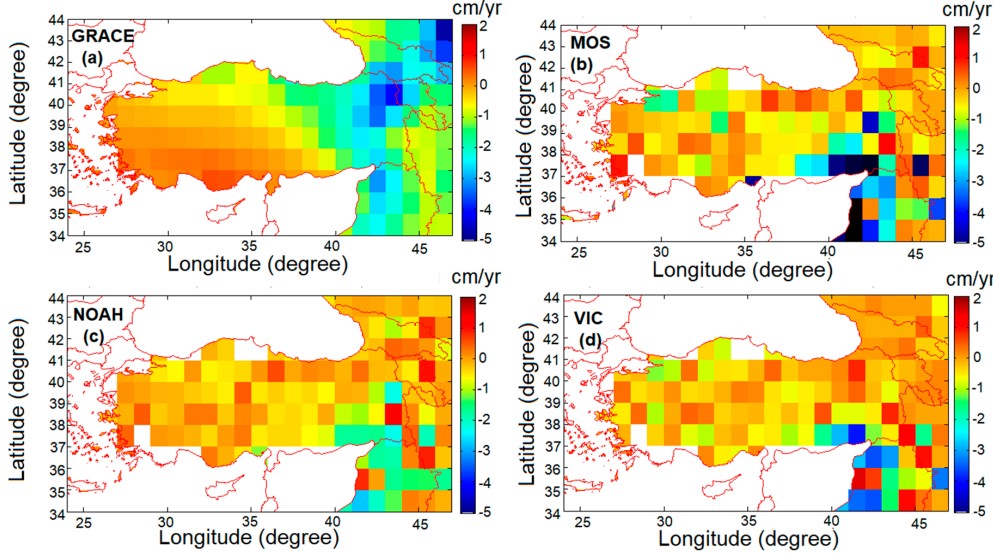

**Figure 5.** Linear trend of TWS variation of Turkey according to (**a**) GRACE data and GLDAS (cm/yr) models, (**b**) MOS, (**c**) NOAH, and (**d**) VIC concerning the 2002–2016 period.

Trend plots of Turkey according to GRACE data (Figure 5a) also show lateral increasing variation from the eastern part to western part, as observed in phase plots (Figure 4a), while GLDAS models

plots also show lateral variation, but in a mostly heterogeneous way (Figure 5b). Distinctive trend values (≥1 cm/yr) appear in the eastern part of Turkey (Figure 5a,b), in accordance with amplitude values (Figure 3a,d).

Even GRACE and GLDAS 2D plots agree with each other and show the spatial distribution of TWS; however, they do not explain the reason for the above-presented spatial distribution. To study the possible reasons for this, GRACE 2D plots are compared with the TRMM-derived precipitation data shown here as 2D sections. Figure 6 shows the (a1) annual amplitude (cm) of TWS variation from GRACE data and (b1) annual amplitude of precipitation estimated from TRMM model, (a2) annual phase of TWS (degree) estimated from GRACE data and (b2) annual phase of precipitation (degree) estimated from TRMM model, (a3) annual trend of TWS variation from GRACE data (cm/yr), (b3) annual trend of precipitation (cm/yr) estimated from the TRMM model between 2002–2016 in Turkey, and (c1) in-situ meteorological rainfall data.

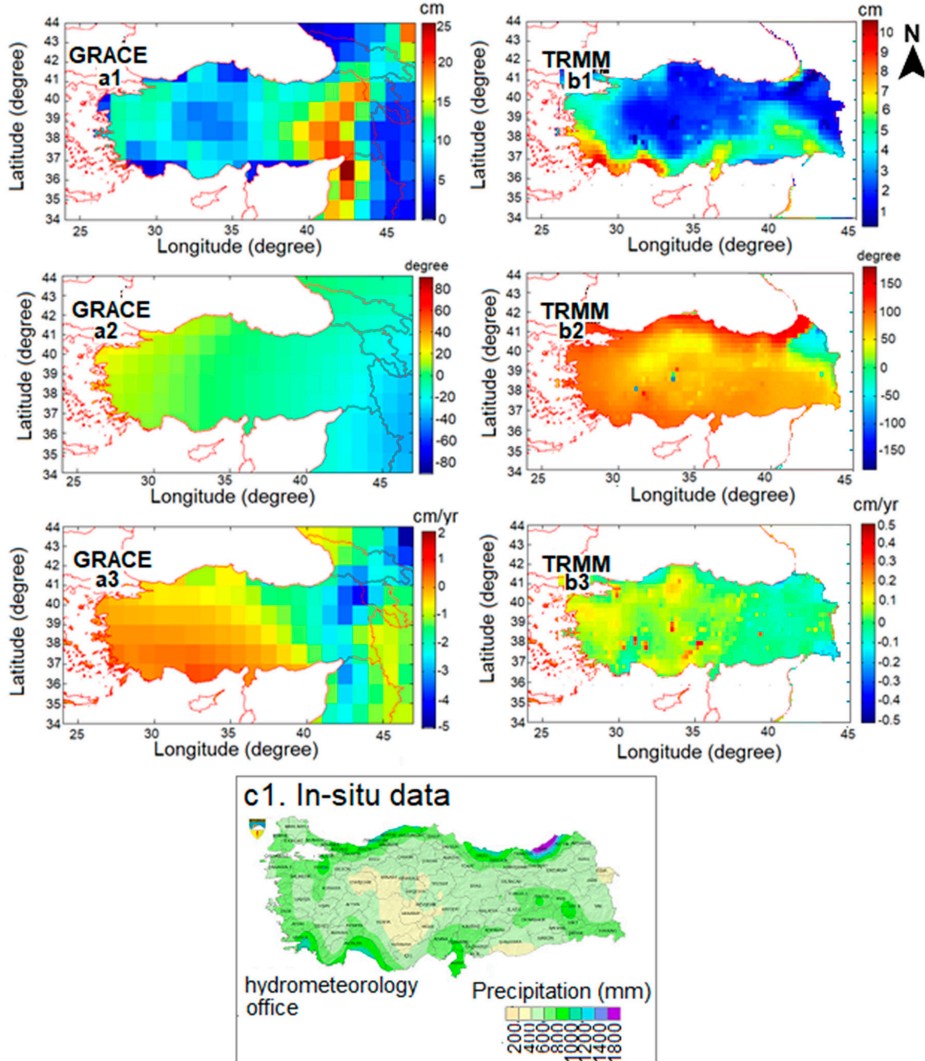

**Figure 6.** (**a1**) Annual amplitude (cm) of TWS variation from GRACE data and (**b1**) annual amplitude of precipitation estimated from TRMM model (scales are different), (**a2**) annual phase of TWS (degree) estimated from GRACE data, (**b2**) annual phase of precipitation (degree) estimated from TRMM model, (**a3**) annual trend of TWS variation from GRACE data (cm/yr), and (**b3**) annual trend of precipitation (cm/yr) estimated from TRMM model between 2002–2016 in Turkey. (**c1**) Turkish state meteorological service's annual cumulative rainfall distribution (1981–2010) map produced by kriging the in-situ rainfall data of 255 meteorological stations on a GIS platform. Scales are different.

Figure 6a1,b1 show larger GRACE-TWS (in red, ≥20 cm) and TRMM anomalies (~7 cm) in the Eastern part of Turkey (see also (Figure 3b–d). The smallest TRMM-precipitation anomalies are observed in central Anatolia (in blue, ≤3 cm), in agreement with GRACE TWS (Figure 6a1). In addition, the TRMM 2D amplitude (Figure 6b1) and phase plot (Figure 6b2) reveal the input of the water because of the precipitation observed on the shorelines of Turkey (Black Sea, Aegean, Mediterranean). This result shows agreement with the GLDAS models' TWS annual amplitude plots (Figure 3b–d) and also with the in-situ annual cumulative rainfall distribution map of the Turkish state meteorological service (Figure 6c1). Trend plots in Figure 6a3,b3 indicate a positive trend through the western part, especially to the Aegean and Mediterranean shorelines of Turkey. Previous studies focused on groundwater loss and reservoir/lake storage change [58,59] also support an acceleration signal in GRACE analyses over the western part of Turkey [38], which has been spatially mapped in this paper. This acceleration seems to be related to the precipitation patterns or flow (from the precipitation-rich eastern part to precipitation-lacking Central Anatolia). However, this might not be the only reason for this and has to be studied in more depth (e.g., human-induced groundwater withdrawal, as mentioned in [36]). After studying the spatial mass distribution and its causes, we will now have a general look and try to focus on the analysis of the nature of the long-term mass change in TWS in Turkey (Section 3.3).

### *3.3. Long-Term Mass Change*

The long-term variations of TWS are estimated and investigated according to GRACE and GLDAS models (MOS, NOAH, VIC) in Turkey. Figure 7 indicates residual mean monthly TWS variations in Turkey (cm) from GRACE and GLDAS models between 2002 and 2016. The mean TWS values of a specific month are first calculated by taking the average of all grids (77 grids) with different geographic coordinates in Turkey for each month of a specific year (e.g., 2002/04, 2002/05), and then all corresponding months of a specific year are averaged between them (January 2002, January 2003...). According to GRACE data, Figure 7 indicates an increasing behavior of the TWS variation from January to April. The monthly maximum of mean TWS at about 10 cm is reached in April. The TWS variation is negative from April to September, with a minimum of −12.88 cm in September. This corresponds to possible dry summer periods, as is normal for the Mediterranean climate. This decrease is followed by a gradual increase from September to December, when we expect more precipitation or snow for the central and eastern parts of the country. The seasonal variations of TWS estimated from GLDAS models are in good agreement with the results from GRACE.

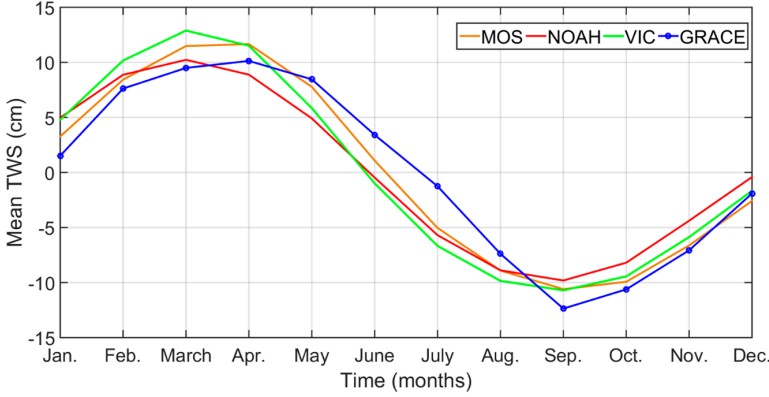

**Figure 7.** Residuals mean monthly TWS changes for Turkey land calculated from GLDAS-MOS (in orange), GLDAS-NOAH (in red), and GLDAS-VIC (in green) models, and from GRACE data (blue circled) for 2002–2016.

In Figure 8, the long-term variation of mean monthly TWS in the spring, summer, autumn, and winter periods from 2002 to 2016 is studied from GRACE-derived TWS data (cm/month). The larger amplitudes of the long-term seasonal TWS signal are observed in spring (max: 20.64 cm, mean:

12 cm), corresponding to April (Figure 7). These orders of amplitudes are followed by the winter period, which is larger than the summer period. Surprisingly, the weaker TWS values (max: 3 cm, mean: −7 cm) are observed in autumn, corresponding to September (Figure 7), instead of summer, as one might expect. This can be explained by the emphasized drought conditions, arising after dry summers (see Section 3.4.1), which are systematically observed in September almost every four years during the studied time period (September/2004, September/2008, October/2014 see also Figure 2). For all seasons, an important decrease is observed in August 2008. This corresponds to the severe drought events experienced in Turkey, especially in 2008 (also Figure 2). To conclude, there is a major decreasing trend close to −1 cm/yr in Turkey for the studied period which has to be taken into account. The exhibition of the negative trend signal of GRACE time series can be attributed to drought conditions and groundwater withdrawal [35] for the years 2003–2008. The proper study of the long-term change of signal can reveal a drought prediction with the amplitude of a decreasing trend (cm/yr), as shown in our results, and can lead to a drought preparedness. In the next section, we will study the parameters (e.g., precipitation, rainfall rate, evapotranspiration, soil moisture) which might have influenced/contributed to the amplitude of the above-presented spatial distribution (Section 3.2) and long-term mass change (Section 3.3).

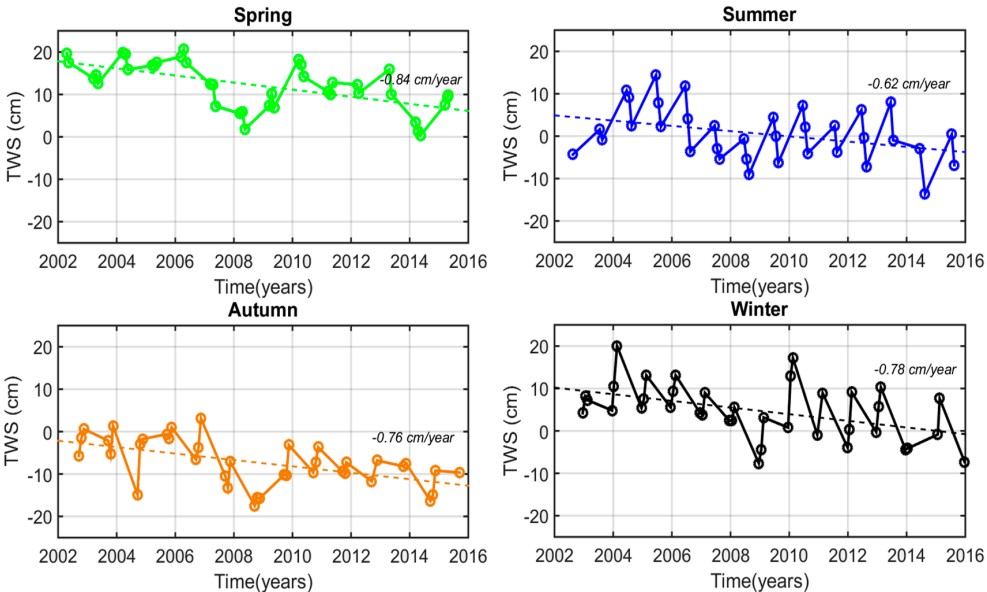

**Figure 8.** Long-term variation of mean monthly TWS of Turkey concerning (**a**) spring, (**b**) summer, (**c**) autumn, and (**d**) winter period from 2002 to 2016.

## 3.4. Impacts on TWS Variations

In this section, we performed statistical analysis with IBM SPSS25 software in order to understand descriptive measures and correlations between studied variables and GRACE TWS.

### 3.4.1. Impact of Precipitation

In order to understand the impact on TWS variations, we used estimated precipitation (cm/month) from TRMM, which is an average rate over a month, and compared the result to the GRACE data. The comparison is shown in Figure 9, which is concerned with the mean monthly precipitation from the TRMM model and the mean monthly TWS derived from GRACE. The TRMM-derived precipitation ranges from 0.71 (July 2015) to 15.36 cm/month (December 2012), while the GRACE-derived TWS ranges from −17.48 (August 2008) to 20.64 cm (April 2006). The amplitude of the TRMM is smaller compared to GRACE data. This can be explained by the fact that GRACE data, displaying more of

an increasing and decreasing trend, is not only affected by the precipitation, but also by the other parameters, as we further investigate in this paper.

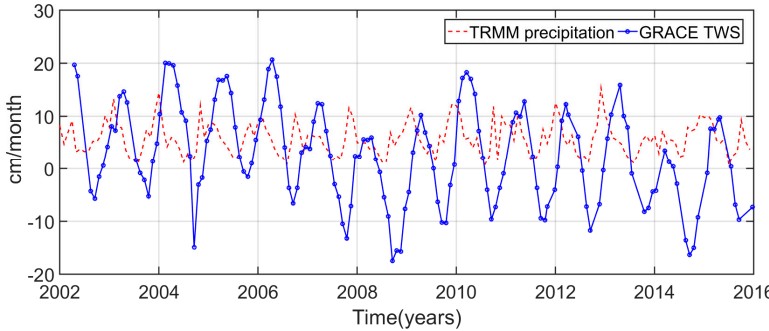

**Figure 9.** Mean monthly precipitation (cm/month) from TRMM model (in red) and mean monthly GRACE TWS (cm, in blue with circle) of Turkey during 2002–2016.

To give numerical orders of magnitude concerning the correlation between TRMM and GRACE TWS time series, firstly, with the Kolmogorov-Smirnov method, the data is checked to assess whether the distribution is normal or not. The results show that the TRMM distribution is not normal ($p = 0.036 < 0.05$). Hence, Spearman's rho method is preferred to more appropriately study the correlation between variables. As seen Table 3, there is a significantly positive correlation between TRMM and GRACE TWS with 0.34.

**Table 3.** Correlation between TRMM precipitation and GRACE TWS according to Spearman's rho method.

| **Correlations** | | | | |
|---|---|---|---|---|
| | | | **TRMM** | **GRACE** |
| Spearman's rho | TRMM precipitation | Correlation Coefficient | 1.000 | 0.337 ** |
| | | Sig. (2-tailed) | | 0.000 |
| | GRACE TWS | Correlation Coefficient | 0.337 ** | 1.000 |
| | | Sig. (2-tailed) | 0.000 | |

** Correlation is significant at the 0.01 level (2-tailed).

### 3.4.2. Impact of Rainfall Rate

Rainfall rate values ($kg/m^2$) are available in the GLDAS models data files. We extracted corresponding values in the studied region, and converted (cm/month), averaged, and calculated the residual. Figure 10 shows the residual mean monthly rainfall rate (cm/month) extracted from GLDAS models (MOS, NOAH, VIC), from TRMM and residual mean monthly TWS variation from GRACE data from 2002 to 2016 for Turkey. The rainfall data changes between $-3.5$ and $6.5$ cm/month (MOS) and $-5$ and $9.5$ cm/month (TRMM). In Figure 10, a good agreement between all models of GLDAS (MOS, NOAH, VIC) and TRMM data is observed. The correlation between GRACE TWS, TRMM precipitation, and GLDAS models' rainfall rate is given in the Table 4.

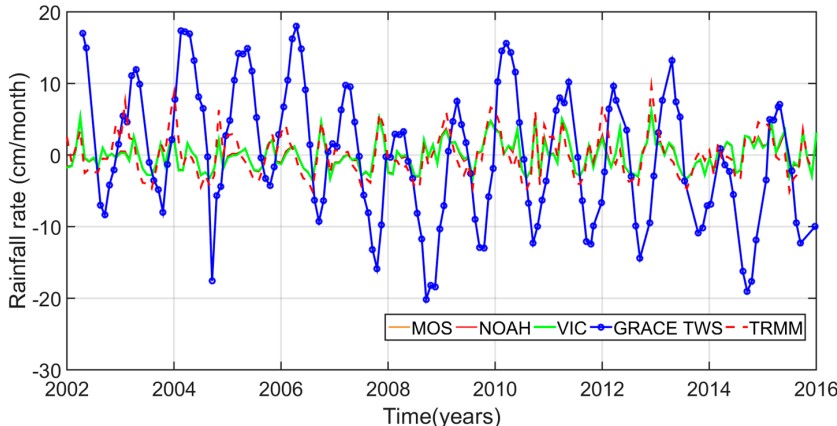

**Figure 10.** Residual mean monthly rainfall rate (cm/month) from GLDAS-MOS (in orange), GLDAS-NOAH (in red), and GLDAS-VIC (in green) models, and from TRMM (in red striped) and residual mean monthly TWS variation from GRACE data (in blue with circle) from 2002 to 2016 for Turkey.

**Table 4.** Correlation between GRACE TWS, TRMM precipitation, and GLDAS models (MOS, NOAH, VIC) rainfall rate according to Spearman's rho method.

| Correlations | | | GRACE | TRMM |
|---|---|---|---|---|
| Spearman's rho | GRACE TWS | Correlation Coefficient | 1.000 | 0.338 ** |
| | | Sig. (2-tailed) | | 0.000 |
| | MOS rainfall rate | Correlation Coefficient | 0.243 ** | 0.826 ** |
| | | Sig. (2-tailed) | 0.003 | 0.000 |
| | NOAH rainfall rate | Correlation Coefficient | 0.244 ** | 0.826 ** |
| | | Sig. (2-tailed) | 0.003 | 0.000 |
| | VIC rainfall rate | Correlation Coefficient | 0.227 ** | 0.788 ** |
| | | Sig. (2-tailed) | 0.005 | 0.000 |
| | TRMM precipitation | Correlation Coefficient | 0.338 ** | 1.000 |
| | | Sig. (2-tailed) | 0.000 | |

** Correlation is significant at the 0.01 level (2-tailed).

There is a positive correlation between GRACE TWS and the rainfall rate derived from GLDAS models (r = ~0.24) and a strong positive correlation between TRMM precipitation and the rainfall rate derived from GLDAS models (r = ~0.82).

### 3.4.3. Impact of Evapotranspiration

The estimation of evapotranspiration (ET) has been performed by using the water balance equation [60,61] as is traditional for a closed basin area, in the case of available observed streamflow data (see Equation (6)):

$$ET(t) = P(t) - R(t) - TWS_{GRACE} \qquad (6)$$

where $t$ is the time; $TWS_{GRACE}$ is the water storage derived from GRACE data; and $P(t)$, $R(t)$, and $ET(t)$ are the precipitation provided from TRMM, runoff extracted from available GLDAS files of different models, and evapotranspiration, respectively. Figure 11 compares the residual mean monthly evapotranspiration values calculated from Equation (6) and GRACE TWS data in order to understand

the role of the evapotranspiration effect on the TWS parameter. Evapotranspiration values range from −3.8 to 8.6 cm/month (VIC). This finding shows that both data (GLDAS evapotranspiration and GRACE TWS) are in-phase. The amplitudes of evapotranspiration seem to impact the rainfall rate equally to the TWS amplitudes.

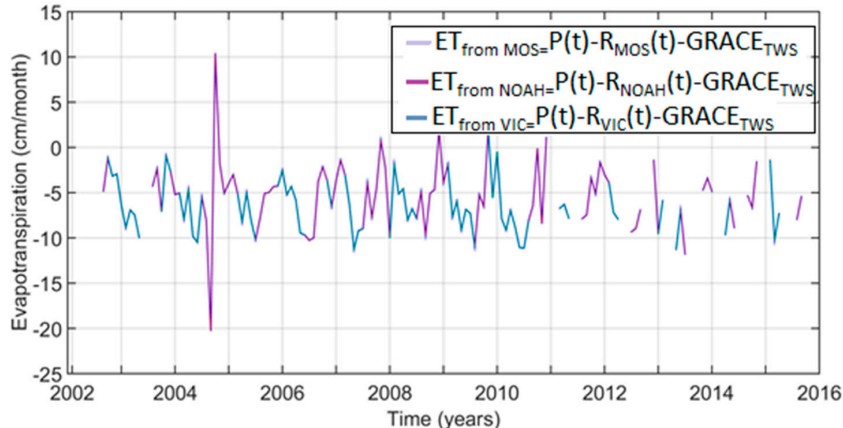

**Figure 11.** Mean monthly evapotranspiration (cm/month) calculated from precipitation (TRMM) minus runoff (from model (MOS, NOAH, VIC)) minus GRACE TWS variations between 2002 and 2016 for Turkey.

### 3.4.4. Impact of Soil Moisture

First, for each GLDAS land model, soil moisture values ($kg/m^2$) available in the GLDAS data files are extracted from the global grids, and then reduced to the studied region scale. In addition, according to the models, soil moisture values are summed with respect to the number of model layers, as follows: three-layer model for MOS and VIC and four-layer model for NOAH are converted to cm and averaged, and the residual mean monthly variations are finally calculated. Figure 12 shows the comparison between residual mean monthly soil moisture variation (cm) obtained from GLDAS models and residual mean monthly GRACE TWS of Turkey from 2002 to 2016. Soil moisture values ranges from −15.4 to 18.9 cm (MOS), while GRACE TWS data ranges from −16 to 20 cm. According to Figure 12, it can be concluded that for Turkey, the most efficient parameter producing the important part of the GRACE TWS signal is the soil moisture. This statement is supported not only by the MOS model, but also by other GLDAS models (NOAH, VIC). Table 5 shows the correlation coefficients between GRACE TWS and soil moisture obtained from GLDAS models (MOS, NOAH, VIC).

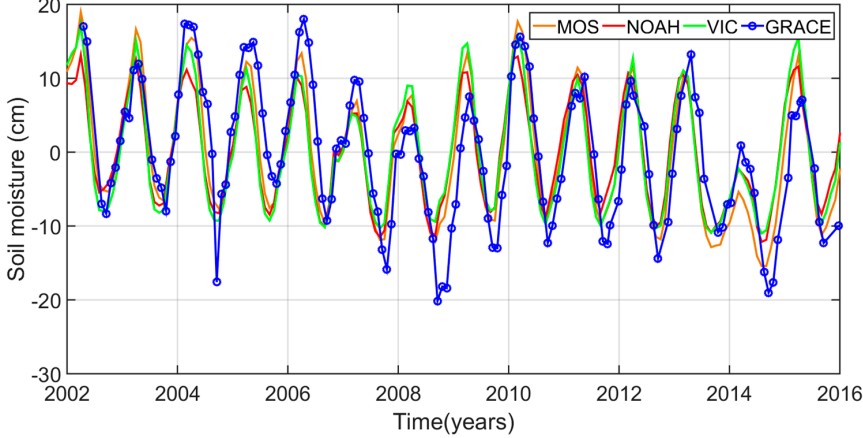

**Figure 12.** Residual mean monthly soil moisture variation (cm) of Turkey from GLDAS MOS (in orange), GLDAS-NOAH (in red), and GLDAS-VIC (in green) models, and residual mean monthly TWS variation from GRACE data (in blue with circle) from 2002 to 2016.

**Table 5.** Correlation between GRACE TWS and soil moisture derived from GLDAS models (MOS, NOAH, VIC) according to Spearman's rho method.

| Correlations | | | |
|---|---|---|---|
| | | | **GRACE** |
| Spearman's rho | GRACE TWS | Correlation Coefficient | 1.000 |
| | | Sig. (2-tailed) | |
| | MOS soil moisture | Correlation Coefficient | 0.840 ** |
| | | Sig. (2-tailed) | 0.000 |
| | NOAH soil moisture | Correlation Coefficient | 0.797 ** |
| | | Sig. (2-tailed) | 0.000 |
| | VIC soil moisture | Correlation Coefficient | 0.790 ** |
| | | Sig. (2-tailed) | 0.000 |

** Correlation is significant at the 0.01 level (2-tailed).

The correlation coefficient between GRACE TWS and soil moisture derived from the MOS model with r = 0.84 indicates a strong positive relation between these two times series and supports that the GRACE TWS signal is mostly dependent on soil moisture content in the studied region. To sum up, soil moisture is a key parameter in terms of drought monitoring because, as mentioned previously in Section 3.1, the insufficiency of soil moisture is a turning point indicating the change from classical drought, observed because of the lack of precipitation, to the agricultural drought. GRACE TWS time series are very sensible to agricultural drought (2006–2008).

*3.5. Understanding the Drought and Its Relation with Climatic Change*

In Section 3.4, we investigated the impact and correlation of different parameters (e.g., precipitation, soil moisture, etc.) on the signal amplitude, specifically on the TWS time series. We found that the precipitation is an important parameter which governs the pattern of spatial mass distribution (Section 3.2) and soil moisture produces the most important part of the GRACE TWS signal (Section 3.4.4). In this part, we decided to combine the available data and our findings with the self-calibrating Palmer Drought Severity Index, ENSO, and NOA index, which use also precipitation, temperature, soil moisture, and so on [51,52] to understand the type, the variability, and the severity of a drought event.

3.5.1. Self-Calibrating Drought Severity Index (SCPDSI)

In Figure 13, non-seasonal GRACE TWS data are compared with the SCPDSI drought severity index.

The analysis of the SCPDSI time series shows a good agreement in terms of the increasing and decreasing trend of the signal with GRACE-derived residual TWS. From a general point of view, in Figure 13, except for some small variations, SCPDSI time series show that progressive decreasing periods (2002–2008, 2010–2014) and increasing periods are (2008–2010, 2014–2016) followed up with the GRACE TWS. Nevertheless, the SCPDSI index is not as sensitive as GRACE data to small variation, especially during 2002–2006 (miss the increase 02/2003–02/2004). September 2004 (may be also an overestimation of GRACE solution), 2008, and October 2014 appear as dramatic drought cases for both time series.

Statistically speaking, according to the Kolmogorov-Smirnov method, p is equal to 0.2 and the Pearson Correlation can be used here to study the correlation coefficients between GRACE TWS and the self-calibrating Palmer Drought Severity Index (SCPDSI), as shown in Table 6.

A positive correlation is found to be r = 0.25. To sum up, according to GRACE data, the SCPDSI index, and historical data, the reason for the drought can be primarily categorized as meteorological. In this step, we decided to extend our results and to more deeply study the underlying processes of the drought event, i.e., the causes resulting in a lack of precipitation from the point of view of climatic

change. For this reason, in the next section (Section 3.5.2), GRACE TWS anomalies are compared with the ENSO and NAO index, improving our understanding of the occurrence of drought events.

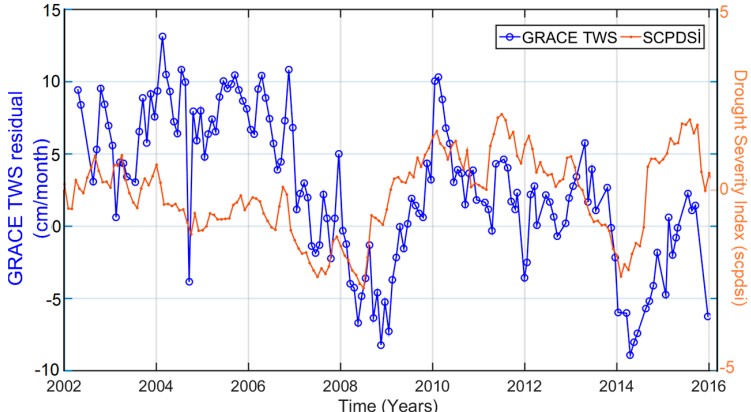

**Figure 13.** Mean monthly residual TWS variation from GRACE data (blue circled line, seasonal signal removed) and mean monthly self-calibrating Palmer Drought Severity Index (orange dotted line) from ~2002 to 2016 of Turkey.

**Table 6.** Correlation between residual GRACE TWS and self-calibrating Palmer Drought Severity Index (SCPDSI).

| Correlations | | | |
|---|---|---|---|
| | | **SCPDSI** | **GRACE TWS** |
| SCPDSI | Pearson Correlation<br>Sig. (2-tailed) | 1 | 0.246 **<br>0.002 |
| GRACE TWS | Pearson Correlation<br>Sig. (2-tailed) | 0.246 **<br>0.002 | 1 |

** Correlation is significant at the 0.01 level (2-tailed).

### 3.5.2. El Niño Southern Oscillation (ENSO) and North Atlantic Oscillation (NAO) Indices

Figure 14 compares the SST3.4 and NAO time series with the non-seasonal GRACE TWS anomaly time series.

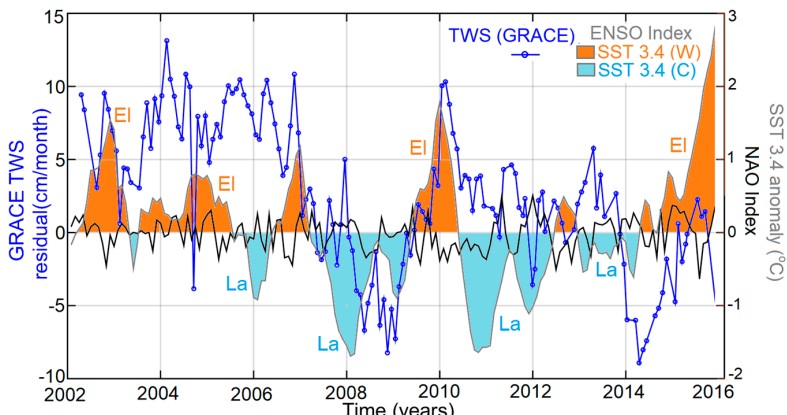

**Figure 14.** Mean monthly residual TWS variation from GRACE data (blue circled line, seasonal signal removed) compared with the ENSO index Niño SST3.4 (El: El Niño and La: La Niña; Warm: orange shading and Cool: cyan shading) Niño SST3.4. (prepared as in [62]) and with monthly mean NAO index (in black).

According to [63], ENSO also affects the Mediterranean winter climate. During El Niño events, the Mediterranean cyclone track is shifted northward, which affects precipitation. Moreover, less precipitation in southwestern Europe, as well as the Black Sea area, during cold events, but more precipitation in the same regions during warm events are founded [64]. Figure 14 reveals that cold events (La Nina: cyan shading) corresponding to La Nina effect occur between 2006–2009 and 2010–2014. This can be interpreted as a climatic impact creating a lack of precipitation in Turkey. This statement is also validated by the decreasing trend of GRACE TWS and SCPDSI (Figure 13) time series. This lack of precipitation due to the possible effect of the La Nina phase first results in a meteorological drought (Figure 13), which then turns into agricultural and hydrological drought, as mentioned in [55]. Spatially speaking, as a possible proof of the impact of ENSO, as mentioned in [64], Black sea coasts in Figure 6b1 show a low (≤3 cm) annual amplitude of precipitation estimated from the TRMM model, unlike the other coasts.

According to Figure 14, the warm phase El Nino (orange shading) is observed between 2002 and the beginning of 2007 (with some interaction of the cold phase creating instantaneous drops) and 2014–2016. The climatic impact of the warm phase, resulting in an increase in precipitation, seems to appear as the increase of GRACE TWS and ENSO, while SCPDSI is more sensitive to the decrease of signal amplitude in this period due to the interaction of the cold phase (Figure 13). GRACE TWS anomalies (Figure 14) and rainfall rate (Figure 10) are important for these time intervals cited above. Spatially speaking, the bigger annual amplitudes of precipitation (≥7 cm) estimated from the TRMM model (Figure 6b1) are observed in Southeastern, Mediterranean, and Aegean parts of Turkey.

Concerning the NAO index, the time series do not show any strong positive or negative anomalies or trends. As a reminder, strong positive phases (+) tend to be associated with below-average precipitation, while strong negative phases (-) are related with the above average temperature and precipitation anomalies. In this case, the amplitude of the NAO time series is small. It can be concluded that NAO does not have any significative impact on the studied area. Table 7 shows the correlation between GRACE TWS with ENSO and the NAO index. There is a positive correlation in the order of r = 0.3 between GRACE TWS and the ENSO index. Additionally, GRACE TWS and the NAO index show a small negative correlation (r = −0.05).

**Table 7.** Correlation between residual GRACE TWS and ENSO Index.

| Correlation | | |
|---|---|---|
| | | **GRACE TWS** |
| GRACE TWS | Pearson Correlation | 1.000 |
| | Sig. (2-tailed) | |
| ENSO index | Pearson Correlation | 0.295 ** |
| | Sig. (2-tailed) | 0.000 |
| NAO index | Pearson Correlation | −0.049 |
| | Sig. (2-tailed) | 0.552 |

** Correlation is significant at the 0.01 level (2-tailed).

## 4. Discussion

Our results provide a broad context for the current hydrological status in Turkey by combining various external data sets (e.g., hydrological models, remote sensing techniques, drought indices) and reveal new drought events, spatial extension of the mass change, long-term variation, impacts on TWS, and the effect of climatic change, in addition to the previous studies within the GRACE mission operation time. The limitations of the work are related to the spatial resolution of the GRACE mission, which does not allow monitoring of the very high resolution surface mass change (e.g., dams, reservoirs); the lack or inaccessibility of the in-situ rainfall data. Which could provide information about groundwater withdrawal; and additionally and more specifically, the limited satellite mission

lifetime that ended in 2017. Even though there has been a new following mission, "GRACE-FO", which started operating on 22 May, there is a data gap between the two missions. For this reason, as future research to generate a new perspective to drought analysis, there is the aim to develop statistical modelling from GRACE time series. This connection between GRACE and GRACE-FO revealed by statistical modelling will be valuable in terms of the continuity of drought monitoring and prediction, especially in the case of missing data within the GRACE-FO operation period. We also demonstrated that GRACE is more sensitive to agricultural and hydrological drought and less sensitive to meteorological drought, which occurs in the case of a lack of precipitation, increase of temperature, and decrease in humidity. GRACE data might be combined with the datasets derived from remote sensing techniques that measure above-cited external data sets to conduct a more sensitive analysis and to predict meteorological droughts [65]. The combined solutions can be assessed in different regions to test the sensibility of the GRACE data to differentiate different types/states of drought.

## 5. Conclusions

According to the drought analysis studied in this paper from GRACE-derived TWS time series, Turkey experienced dramatic drought events in 09/2004 (may also be an overestimation of GRACE solution), 09/2008, and 10/2014. Moreover, TWS decreasing periods are recorded as follows: 04/2002–09/2004; 02/2006–09/2008; 03/2010–10/2014. In terms of assessment of the drought, GRACE can help to better predict the possible drought (starting from 02/2006) nine months before, with a decreasing trend observed in GRACE TWS times series compared to previous studies which do not take satellite gravity data (see only since 11/2006) into account. Moreover, the GRACE signal is more sensitive to agricultural and hydrological drought compared to meteorological drought. Spatially, mass amplitudes are larger (20 cm) in the eastern part compared to central (2 cm, smaller) or Aegean parts, related to received precipitation levels. Shorelines also show distinctive values compared to the central part. There is an acceleration signal from the eastern side to the western side, which is related to the precipitation. Concerning long-term mass change, Turkey experiences a decreasing trend in the order of 1 cm/yr. Rainfall rate, evapotranspiration, and precipitation constitute a small part of the signal, while soil moisture is the parameter most affecting the GRACE signal in the studied region according to soil moisture values derived from GLDAS models and GRACE TWS results having a strong correlation (r = 0.84). Precipitation has a specific impact on the pattern of the spatial mass distribution. In Turkey, we observed a meteorological drought turning into agricultural and hydrological drought due to the climatic impact of the La Nina effect (cold phase) resulting in a lack of precipitation in Turkey. The GRACE signal is very sensitive to this climatic change. It is worth mentioning that the NAO index does not show any meaningful anomalies and correlation with GRACE TWS (r = −0.05). Finally, in order to real-time monitor and estimate possible drought conditions in the future, either in Turkey or in another region, we propose the combination of the new and up-to-date satellite gravity mission data (GRACE-FO), offering more accurate measurements and providing information about mass decreasing and increasing trends; the precipitation to understand the spatial mass distribution patterns; soil moisture data (models and also in-situ) to monitor the occurrence of a possible agricultural drought; the ENSO index to predict possible excess or deficiency in precipitation; and the drought indices, which provide information about the type and the variability of the drought.

**Author Contributions:** Data curation, G.O.A.; Funding acquisition, S.J.; Methodology, G.O.A. and S.J.; Resources, G.O.A.; Software, G.O.A.; Supervision, S.J.; Writing – original draft, G.O.A.; Writing – review & editing, G.O.A. and S.J.

**Funding:** The work is supported by the Strategic Priority Research Program of Chinese Academy of Sciences (Grant No. XDA23040102) and Startup Foundation for Introducing Talent of NUIST (Grant No. 2243141801036).

**Acknowledgments:** We thank the following organizations for providing the data used in this work: the GRACE Project and CSR (Center for Space Research, Univ. Texas), the TRMM projects, and a Global Land Data Assimilation System (GLDAS) developed jointly by scientists at the National Aeronautics and Space Administration (NASA) Goddard Space Flight Center (GSFC) and the National Oceanic and Atmospheric Administration (NOAA) National Centers for Environmental Prediction (NCEP). We are very thankful to three anonymous reviewers who

helped us in improving the quality of our manuscript and also to Assist. Kamil Teke, Assoc. Prof. Semra Türkan from Hacettepe University, and Mustafa Serkan Işık from Istanbul Technical University for their help.

**Conflicts of Interest:** The authors declare no conflict of interest.

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
