# Peer review of "Hydrologic Mass Changes and Their Implications in Mediterranean-Climate Turkey from GRACE Measurements"

_remotesensing, doi:10.3390/rs11020120_

Reviewer 1 Report

Overall estimate: The paper is interesting and could be published after major review. The authors should consider the following points:

In the abstract some more robust results should be included.

In line 16 as well in the  line 458 authors should mention the specific parameters.

In line 32 the word “these” should be omitted.

In the introduction section, based on literature, more information about the use of the appropriate tools (remote sensing technique or/and hydrological modeling approaches) to define the terrestrial water storage at different special scales should be mentioned.

I strongly recommend a flow chart to be presented in the methodology section.

For comparison reasons, the results should be presented at the same grid scale (see for instance figure 6).  

The presentation of Figure 11 is very confusing and unclear. 

Extensive discussion must be included regarding the limitations and usefulness of the proposed approach.

Author Response

Dear Reviewer, we would like to thank you for the time that you have accorded for reviewing our manuscript and we are grateful for your comments and corrections which will help to produce a better research paper. We invite you to find our answers and corrections to your comments and suggestions as below. We hope that our revisions will be satisfactory by your side.

1.     Reviewer #1 wrote: In the abstract some more robust results should be included.

We added some more robust results in the abstract as suggested. Please see below corresponding new findings. GRACE can help in better predict the possible drought 9 months before in terms of a decreasing trend compared to previous studies which doesn’t take account satellite gravimetric data. Moreover, GRACE signal is more sensitive to agricultural and hydrological drought compared to meteorological drought.

2.     Reviewer #1 wrote: In line 16 as well in the line 458 authors should mention the specific parameters.

We have added rainfall rate, evapotranspiration, soil moisture to the corresponding lines.

3.     Reviewer #1 wrote: In line 32 the word “these” should be omitted.

“These” has been omitted.

4.   Reviewer #1 wrote: In the introduction section, based on literature, more information about the use of the appropriate tools (remote sensing technique or/and hydrological modeling approaches) to define the terrestrial water storage at different special scales should be mentioned.

The introduction has been modified as recommended by reviewer.  

5.   Reviewer #1 wrote: I strongly recommend a flow chart to be presented in the methodology section.

We prepared a table summarizing studied research topics, used input data, methodology and additional applied processing in this paper at the end of the data and methods section before presenting results.

6.   Reviewer #1 wrote: For comparison reasons, the results should be presented at the same grid scale (see for instance figure 6).  

We have corrected the figure 6 and put it in the same scale. Also we added the in-situ meteorological data as recommended by another reviewer. We could access to the database of the Turkish state meteorological service which releases spatially annual cumulative rainfall distribution (1981-2010) map produced by kriging the rainfall data of 255 meteorological stations on a GIS platform. In-situ data and TRMM data are correlated. We added this new in-situ map to the figure 6 to compare with GRACE and TRMM results.

7.   Reviewer #1 wrote: The presentation of Figure 11 is very confusing and unclear. 

The figure 11 has been corrected.

8.   Reviewer #1 wrote: Extensive discussion must be included regarding the limitations and usefulness of the proposed approach.

Thank your comments very much. In the revised manuscript, we have added some discussions on broadest context possible and limitations as well as reviews of previous studies.  Also we have added some discussions on future research direction or problems are presented and discussed.

Reviewer 2 Report

The paper presents research on the correlation of TWS variation obtained from GRACE satellite, TRIMM and other dataset and compared with the ENSO climate index.   

The paper is properly addressed and accurate but some more analysis should be necessary to make more robust some assessments. 

More details in the attached PDF file 

Author Response

Dear Reviewer, we would like to thank you for the time that you have accorded for reviewing our manuscript and we are grateful for your comments and corrections which will help to produce a better research paper. We invite you to find our answers and corrections to your comments and suggestions as below. We hope that our revisions will be satisfactory by your side.

Point 1: Reviewer #2 wrote: May be useful if you also consider the correlation with other climate index like NAO.  

Response 1: Thank you very much for this suggestion. I am sure that there is an interesting link and effect between NAO and Turkish climate change. However, the starting point and the scope of this paper wasn’t studying the climatic change and its effect on Turkey. However, to understand the lack of precipitation and the main raisons of the drought, our results lead us to study as a beginner also ENSO. May be, it will be more appropriate to just study the climatic effects and its impact on Turkey and then it will be more interesting to mention about all this kind of climate index.

Point 2: Reviewer #2 wrote: It would be interesting add some comparison/validation between TRIMM data and rainfall from ground meteorological stations.

Response 2: We could only access to the database of the Turkish state meteorological service which releases spatially annual cumulative rainfall distribution (1981-2010) map produced by kriging the rainfall data of 255 meteorological stations on a GIS platform. In-situ data and TRMM data are correlated. We added this new in-situ map to the figure 6 to compare with GRACE and TRMM results.

Point 3: Reviewer #2 wrote:  Line 460: “..Turkey observe dramatic drought events in 09/2004..” Is not clear if drought events come from Palmer index analysis or from TWS data. In addition, the point 09/2004 in the time series of GRACE TWS seems an out-layer do you consider could be an error?

Response 3: To be more clear we modified the sentences as follows: According to drought analysis studied from GRACE derived TWS time series, Turkey observe dramatic drought events in 09/2004, 09/2008 and 10/2014. Thank you for your question. Concerning the point 09/2004 we think that September, especially every 4 years (e.g. 2004, 2008), is an important period which show decreasing behavior. This is shown not only in figure 13 but also in Figure 7 and Figure 8.

Point 4: Reviewer #2 wrote:  Figure 11 is not clear, there are some errors in the legend or in the chart.

Response 4: Figure 11 has been corrected.

Point 5: Reviewer #2 wrote:  In Calò et al 2017 there are also described the effect of drought and water extraction on groundwater and TWS in the central Turkey. Also the groundwater storage can affect the TWS.

Response 5: Thank you for this good reference. The observed drought in Turkey lead to water extraction on groundwater for agricultural purposes. [35] mentioned the effect of drought and the resulting water extraction on groundwater between 2003-2008. May be we can put this reference also as it is so recent but I could not find the correct citation. We would be appreciating if you could give us the entire citation of the above mentioned paper.

[35]: Lenk, O. Satellite based estimates of terrestrial water storage variations in Turkey. J. Geodyn. 2013, 67, 106–110, doi:10.1016/j.jog.2012.04.010.

Point 6: Reviewer #2 wrote:  Do you consider if there also some effects on TWS related to the built of new dam reservoirs or they are almost irrelevant?

Response 6: Thank you very much for this nice question. For this study, we haven’t studied or interpreted the effect of new dam reservoirs so we don’t have further information. However, we know that although reservoirs may cover an important area, their spatial extent may be unresolved by GRACE as mentioned in Longuevergne, Laurent, et al. "GRACE water storage estimates for the Middle East and other regions with significant reservoir and lake storage." Hydrology and Earth System Sciences 17.12 (2013): 4817-4830. Researchers studies surface water reservoirs using a priori information on reservoir storage from radar altimetry. However, this wasn’t the scope of this study.

Minor issues:

Point7: Reviewer #2 wrote:  Reference to figures and sections not require bold fonts

Response 7: We canceled bold fonts.

Point 8: Reviewer #2 wrote:  Line 287 and line 458 remove “...” and add the name of the parameters

Response 8: We included other parameters as follows: some parameters (e.g. precipitation, rainfall rate, evapotranspiration, soil moisture)

Point 9: Please check if all the links to data repository are updated and working.

Response 9: We checked the links and we found that “https://disc.sci.gsfc.nasa.gov/services/grads-gds/gldas”

Does not work. We changed this link with “https://disc.gsfc.nasa.gov/datasets?keywords=gldas&page=1”

Reviewer 3 Report

This manuscript describes the comparison between the GRACE TWS signal and several global land surface models (MOS, NOAH, VIC), including comparison to a teleconnection oscillation index (ENSO) and Palmer Drought Severity Index.

 A lot of work has been put in this study, that is obvious and can be seen from the overall good quality of the figures.

The conclusion should probably be that GRACE can help in better assessment of drougths, but the quality is paper is poor, so that conclusion does not come across very well. 

I advice the authors to better structure their paper, point out what the novelty in the paper is, have someone check the English, and then re-submit their study.

My main remarks: 

In general:

First of all, the English is poor. And that is an understatement. That makes it extremely hard to follow the study logic. 

Second, and more importantly, there is no clear structure in the paper. In the Data and Models section only a few of the used models are described, the rest suddenly appears in Results and Analysis. This paper should do with a very structured description of input data and models.

This paper should also have a method section. Because currently, it is not clear what the researchers of this study actually developed themselves, and what was developed in the input data and models.

After these revisions are made, I would like to ask the authors to discuss what data has been used to process GRACE TWS, and if those data are coming from a GLDAS. If GLDAS or any of their input data have been used to process/calibrate GRACE TWS, then any correlations between GLDAS and GRACE TWS would logically be higher, and possible lead to false high correlations. This analysis would bring some more novelty to the study.

Some major remarks detailed:

Section 2. Data and models

Why does this section not include descriptions PDSI?

Have the equations been developed by the authors?

Section 3. Results and Analyses

In results and analyses, suddenly other data appears and other methods to analyse as well. 

p5. line 158-176. This text is a different format. It looks like a figure caption, but it should probably be text. The text is wholly incomprehensible. Sorry, I did not understand any point the authors wanted to make here.

What did the researchers do in this study and what data did they use without any additional processing (leading to the recommendation that this paper should have a clear distinction between input data and methods used).

Fig 2: are the GRACE TWS minima events in 2004 and 2008 actual droughts? 

Hundreds of minor corrections are needed. I did not get to them all, as I lost track of the paper logic. A few suggestions:

- p1. l32: 'these extreme hydrological events'. What events?

- p1, l33: incorrect English ('to characterise regionally the...')

- p1. l35: 'observation', should be plural.

- p1. l39: '[14] revealed that...' is poor writing.

- p1. l39: why are some things in Italic?

- p2. l43: '[17] studies long-term'. Poor writing.

- p2, l52: CSR was not mentioned before, what is it?

- p2, l62: (why is 'see Fig. 1) in bold?

These above errors keep occuring throughout the paper.

- p17, l478: 'lack time' should be 'lag time'

- p17, l479: 'indexes' should be 'indices'

- p17, l480: 'Hereafter we also aim a statistical modelling .... to predict possible drought...'. Has this been introduced: that is new information and does not belong in a conclusion. I suggest to remove that or put in a discussion section that introduces this topic and elaborates on it.

Author Response

Dear Reviewer, we would like to thank you for the time that you have accorded for reviewing our manuscript and we are grateful for your comments and corrections which will help to produce a better research paper. We invite you to find our answers and corrections to your comments and suggestions as below. We hope that our revisions will be satisfactory by your side.

Point 1: Reviewer #3 wrote: First of all, the English is poor. And that is an understatement. That makes it extremely hard to follow the study logic. 

Response 1: The manuscript has been corrected by a native speaker.

Point 2: Reviewer #3 wrote: Second, and more importantly, there is no clear structure in the paper. In the Data and Models section only a few of the used models are described, the rest suddenly appears in Results and Analysis. This paper should do with a very structured description of input data and models. I advise the authors to better structure their paper.

Response 2: To restructure the paper, we moved the SCDPSI and the ENSO sections which appears in the results and analysis sections to the data and methods part. Additionally, as suggested by another reviewer we added a table which summarize the general structure, input data, research topics, methodology, additional processing that we applied to the used data. We hope that will be satisfactory by your side.

Point 3: Reviewer #3 wrote: what the novelty in the paper is

Response 3: Our study combined multiple remote sensing technics and hydrological models data with satellite gravimetry data. With these up-to-date data and methods, we could be able to reveal new drought events, spatial extension of the mass change, long-term variation, impacts on TWS and the effect of climatic change within the GRACE mission operational time. Our results showed us that GRACE can help better predict the possible drought ~9 months before compared to previous studies. Additionally, we found that GRACE signal is more sensitive to agricultural and hydrological drought compared to meteorological drought. These findings are very important to study and to test in different regions to study the limits of GRACE data, especially now GRACE-FO data, in determining different types of drought events by combining external data sets.

Point 4: Reviewer #3 wrote: The conclusion should probably be that GRACE can help in better assessment of droughts, but the quality is paper is poor, so that conclusion does not come across very well. 

Response 4: In conclusion, we added the following phrases which highlight the use and advantage of GRACE concerning the assessment of drought:

In terms of assessment of drought, GRACE can help in better predict the possible drought (starting from 02/2006) 9 months before in terms of a decreasing trend compared to previous studies which doesn’t take satellite gravimetric data (see only since 11/2006) into account.

*As an example of previous studies which doesn’t take satellite gravimetric data is for example: Kapluhan, E. TÜRKİYE’De Kuraklik Ve Kurakliğin Tarim Etki̇si̇. Marmara Coğrafya Derg. 2013, 487–510

Also, the analysis of GRACE TWS time series and the results of previous works revealed that GRACE signal is mostly sensitive to agricultural drought (insufficient soil moisture content) and hydrological drought (significant reduction in winter precipitation) (2006-2008) compared to meteorological drought (little rain combined with increased temperature and lower humidity) (2012-2014) for the studied region. We added this remark in abstract, in section 3.1, in section 3.4.4 and as well as in conclusion part.

Point 5: Reviewer #3 wrote: This paper should also have a method section. Because currently, it is not clear what the researchers of this study actually developed themselves, and what was developed in the input data and models.

Response 5: We changed the title of the second sections as data and methods instead of data and models. Because GLDAS models actually are our input data that we use with some minor additional processing. To be more clear, we prepared a table at the end of the section 2.

Point 6: Reviewer #3 wrote: After these revisions are made, I would like to ask the authors to discuss what data has been used to process GRACE TWS, and if those data are coming from a GLDAS. If GLDAS or any of their input data have been used to process/calibrate GRACE TWS, then any correlations between GLDAS and GRACE TWS would logically be higher, and possible lead to false high correlations. This analysis would bring some more novelty to the study.

Response 6: To process GRACE data, we developed a Matlab code. Than we could extract the corresponding land grids on Turkey and interested time interval, to perform scaling of coefficients, leakage error correction and also harmonic analysis by least squares adjustment. No ready to use software has been used for this study. GRACE provide TWS, while GLDAS provide TWS independently. The process of GLDAS and GRACE has been performed separately so the positive correlation is meaningful.

 Some major remarks detailed:

Point 7: Reviewer #3 wrote: Section 2. Data and models: Why does this section not include descriptions PDSI?

Response 7: You are right. We move SCPDSI part to the section 2 where we present all used data.

Point 8: Reviewer #3 wrote: Have the equations been developed by the authors?

Response 8: Yes, please see Jin et al. 2013. Jin, S.G., T. van Dam, and S. Wdowinski (2013), Observing and understanding the Earth system variations from space geodesy, J. Geodyn., 72, 1-10, doi: 10.1016/j.jog.2013.08.001. and Jin, S.; Park, J.U.; Cho, J.H.; Park, P.H. Seasonal variability of GPS-derived zenith tropospheric delay (1994-2006) and climate implications. J. Geophys. Res. Atmos. 2007, 112, doi:10.1029/2006JD007772.

Point 9: Reviewer #3 wrote: Section 3. Results and Analyses: In results and analyses, suddenly other data appears and other methods to analyse as well. 

Response 9: You are right. We remarked this fact and we moved the PDSI and ENSO data to the data and methods section.

Point 10: Reviewer #3 wrote: p5. line 158-176. This text is a different format. It looks like a figure caption, but it should probably be text. The text is wholly incomprehensible. Sorry, I did not understand any point the authors wanted to make here.

Response 10: The font size has been changed from 9 to 10 as these lines are interpreting the figure 2. This is not a figure caption.

Point 11: Reviewer #3 wrote: What did the researchers do in this study and what data did they use without any additional processing (leading to the recommendation that this paper should have a clear distinction between input data and methods used).

Response 11: The data used in this study:

ENSO: monthly ERSSTv5 (1981-2010 base period) Niño 1+2 (0-10°South) (90°West-80°West) Niño 3 (5°North-5°South) (150°West-90°West) Niño 4 (5°North-5°South) (160°East-150°West) Niño 3.4 (5°North-5°South) (170-120°West) containing data from 1950 to 2018. We extracted related time interval.

SCPDSI: global land data covering the time period from 1901 to 2016 with a 0.5° latitude-longitude spatial resolution. Then, grids corresponding in Turkey land and the time period of interest from 2002 to 2016 are extracted from global data.

In order to be more clear we added Table 1 which resume all used data, our methodology and additional processing that we performed as requested by another reviewer.

In general, all input data are global. Firstly, we have extracted corresponding grids for Turkey and corresponding time interval (GRACE, GLDAS, TRMM, SCPDSI, ENSO). Concerning GRACE data, as additional processing we performed scale factor and leakage error correction. Concerning GLDAS models data sets we extracted soil moisture (SM), snow moisture equivalent (SWE) and canopy water storage values (ΔCWS). Soil moisture values are summed with respect to the number of models layers: three-layer model for MOS and VIC and four-layer model for NOAH. Than we calculated TWS(GLDAS)=ΔSM+ΔSWE+ΔCWS for each GLDAS model. Besides, for section 3.2 we performed a harmonic analyses and least squares fitting according to [56] for both GRACE and GLDAS data. Annual amplitude (cm), phase (degree) and trend (cm/yr) values are derived. No ready to use software has been used for this study. To perform all above-cited processing, we developed our own Matlab codes.

[56]: Jin, S.; Park, J.U.; Cho, J.H.; Park, P.H. Seasonal variability of GPS-derived zenith tropospheric delay (1994-2006) and climate implications. J. Geophys. Res. Atmos. 2007, 112, doi:10.1029/2006JD007772.

Point 12: Reviewer #3 wrote: Fig 2: are the GRACE TWS minima events in 2004 and 2008 actual droughts? 

Response 12: According to [35], [37], [40], [41], [42], [43] which mention groundwater withdrawal and no snow and no precipitation events in 2008, and validated with GRACE results, we observe a severe drought in 2008. Concerning 09/2004, there is no specific records indicating directly 09/2004 however, in figure 7-8-13 we observe this decrease. We think that September, especially every 4 years (e.g. 2004, 2008), is an important period which show decreasing behavior.

[35] Lenk, O. Satellite based estimates of terrestrial water storage variations in Turkey. J. Geodyn. 2013, 67, 106–110, doi:10.1016/j.jog.2012.04.010.

[37] Yıldırım, Y.Ö. GRACE uydu verileri ile Türkiye’nin uzun dönemli su kütle değişiminin incelenmesi, Gebze Teknik University, 2015.

[40] Türkeş, M. Influence of geopotential heights, cyclone frequency and Southern Oscillation on rainfall variations in Turkey. Int. J. Climatol. 1998, 18, 649–680, doi:10.1002/(SICI)1097-0088(199805)18:6<649::aid-joc269>3.0.CO;2-3.

[41] Türkeş; Serap, A.A.; Zerrin, D. Palmer Kura klık Ġndisi ’ ne Göre Ġç Anadolu Bölgesi ’ nin Konya Bölümü ’ ndeki Kurak Dönemler ve Kuraklık ġiddeti Drought periods and severity over the Konya Sub-region of the Central Anatolia Region according to the Palmer Drought Index. 2009, 7, 129–144.

[42] Kapluhan, E. TÜRKİYE’De Kuraklik Ve Kurakliğin TarimEtki̇si̇. Marmara Coğrafya Derg. 2013, 487–510.

[43] Marım, G.; Şensoy, A.; Şorman, A..; Şorman, A.. Yukarı Fırat Havzası İçin Elde Edilen Kar Çekilme Eğrilerinin Zamansal Analizi ve Modelleme Çalışmaları. In Kar Hidrolojisi Konferansı; Erzurum, 2008.

Point 13: Reviewer #3 wrote:  p1. l32: 'these extreme hydrological events'. What events?

Response 13: “These” has been omitted.

Point 14: Reviewer #3 wrote: p1, l33: incorrect English ('to characterise regionally the...')

Response 14: Corrected.

Point 15: Reviewer #3 wrote: p1. l35: 'observation', should be plural.

Response 15: Corrected.

Point 16: Reviewer #3 wrote: p1. l39: '[14] revealed that...' is poor writing.

Response 16: Corrected.

Point 17: Reviewer #3 wrote: p1. l39: why are some things in Italic?

Response 17: The orders of magnitudes of our results have been written in italic in order to bring out. If it is not appreciated, we can correct the italic style.

Point 18: Reviewer #3 wrote: p2. l43: '[17] studies long-term'. Poor writing.

Response 18: Corrected.

Point 19: Reviewer #3 wrote: p2, l52: CSR was not mentioned before, what is it?

Response 19: Please see line 18 in the abstract where CSR is mentioned. This abbreviation stands for Centre for Space Research (CSR).

Point 20: Reviewer #3 wrote: p2, l62: (why is 'see Fig. 1) in bold?

Response 20: We corrected all bolded references, figures, tables and sections.

Point 21. Reviewer #3 wrote: p17, l478: 'lack time' should be 'lag time'

Response 21: Here, we want to say that Enso helps to understand excess or deficiency in precipitation. We changed lack time with deficiency.

Point 22: Reviewer #3 wrote: p17, l479: 'indexes' should be 'indices'

Response 22: Corrected.

Point 23: Reviewer #3 wrote: p17, l480: 'Hereafter we also aim a statistical modelling .... to predict possible drought...'. Has this been introduced: that is new information and does not belong in a conclusion. I suggest to remove that or put in a discussion section that introduces this topic and elaborates on it.

Response 23: We added a discussion part. Our comment related with the statistical modelling has been removed to the discussion part.

Round  2

Reviewer 1 Report

The revision paper could be accepted for publication in present form

Author Response

Dear Reviewer, 

We really thank you for accepting our paper. This present form has been achieved thanks to your valuable comments. In the revised version of the paper, the English has been rechecked. We hope it will be satisfactory by your side. 

Best regards,

Gonca OKAY AHİ

Reviewer 2 Report

The authors correctly addressed the main comments. Some issues, however, need further control. In the attached file more detailed comments to your reply.

Author Response

Response 1. The NAO index has been added to figure 14 as requested by reviewer 2. We also include an additional part in the second section (Data and methods) concerning NAO. We mentioned about NAO data in the abstract, in the third section, in conclusion and we removed the NAO part from discussion.

Response 2: The resolution of the figure has been improved.

Response 3: Generally, CSR data is more reliable commented also in some research paper (forget about the references) and from our own other research we observed that there is a small difference between CSR and GFZ data but much more with JPL data. That’s why, to be prudent, we keep solutions with CSR data and as you suggest, we added the sentence that the negative peak of 2004/09 may be and over estimation of GRACE solutions (Line 250, 520)

Line 250: However, to be prudent, the decrease in September 2004 may be an over estimation of GRACE solutions.

Line 520: Nevertheless, SCPDSI index are not as sensitive as GRACE data to small variation especially during 2002-2006 (miss the increase 02/2003-02/2004). September 2004 (may be also an overestimation of GRACE solution), 2008 and October 2014 appear as a dramatic drought cases for the both time series.

Response 5: We included the above-cited reference in line 86

More recently, [37] are also described the effect of drought and water extraction on groundwater storage in the central Turkey. They also showed how the groundwater storage can affect the TWS.

Response 6: We changed all the figure where lines corresponding to MOS data in blue to in orange color.
